# Reconstitution of selective HIV-1 RNA packaging in vitro by membrane-bound Gag assemblies

Lars-Anders Carlson[1,2]*, Yun Bai[1†], Sarah C Keane[3,4], Jennifer A Doudna[1,2,5,6], James H Hurley[1,2,6]*

[1]Department of Molecular and Cell Biology, University of California, Berkeley, Berkeley, United States; [2]California Institute for Quantitative Biosciences, University of California, Berkeley, Berkeley, United States; [3]Howard Hughes Medical Institute, Baltimore, United States; [4]Department of Chemistry and Biochemistry, University of Maryland Baltimore County, Baltimore, United States; [5]Howard Hughes Medical Institute, University of California, Berkeley, Baltimore, United States; [6]Molecular Biophysics and Integrated Bioimaging Division, Lawrence Berkeley National Laboratory, Berkeley, United States

**Abstract** HIV-1 Gag selects and packages a dimeric, unspliced viral RNA in the context of a large excess of cytosolic human RNAs. As Gag assembles on the plasma membrane, the HIV-1 genome is enriched relative to cellular RNAs by an unknown mechanism. We used a minimal system consisting of purified RNAs, recombinant HIV-1 Gag and giant unilamellar vesicles to recapitulate the selective packaging of the 5' untranslated region of the HIV-1 genome in the presence of excess competitor RNA. Mutations in the CA-CTD domain of Gag which subtly affect the self-assembly of Gag abrogated RNA selectivity. We further found that tRNA suppresses Gag membrane binding less when Gag has bound viral RNA. The ability of HIV-1 Gag to selectively package its RNA genome and its self-assembly on membranes are thus interdependent on one another.

*For correspondence: lacarlson@berkeley.edu (L-AC); jimhurley@berkeley.edu (JHH)

Present address: †School of Life Science and Technology, ShanghaiTech University, Shanghai, China

Competing interests: The authors declare that no competing interests exist.

## Introduction

Packaging the correct genetic material is a key event in the assembly of an infectious virus. In the case of HIV-1, both the recognition of the genomic RNA, its targeting to the plasma membrane, and assembly of a membrane-enveloped virus around it is carried out by the viral protein Gag (*Rein et al., 2011*; *Sundquist and Kräusslich, 2012*; *Freed, 2015*). The nucleocapsid (NC) domain of Gag has two zinc knuckle motifs which together with adjacent residues interact with the viral RNA (*Berkowitz et al., 1993*; *Lu et al., 2011b*). Mutations in NC that interfere with RNA binding lead to formation of non-infectious viruses (*Aldovini and Young, 1990*; *Gorelick et al., 1990*). Transcription of the integrated HIV-1 provirus results in both spliced and unspliced transcripts, coding for different viral proteins (*Coffin et al., 1997*). Gag selectively packages unspliced, full-length genomic RNA through interactions with its 5' untranslated region (5'UTR), which is different in spliced and unspliced constructs (*Lu et al., 2011b*; *Kuzembayeva et al., 2014*). In addition to sequences that mediate the packaging of the genomic RNA, the 5'UTR also has sequence elements related to transcription processivity (TAR), binding of tRNA[Lys-3] as a primer for reverse transcription (PBS), and genome dimerization (DIS) (*Johnson and Telesnitsky, 2010*; *Lu et al., 2011b*; *Kuzembayeva et al., 2014*).

**eLife digest** HIV-1 is the virus that causes AIDS – short for acquired immune deficiency syndrome – in humans. When HIV-1 infects a person, it targets cells of the immune system, which normally act to defend the body against infections. As the virus spreads from one immune cell to the next, it weakens the immune system so that individuals become more vulnerable to other illnesses. A cell infected with HIV-1 creates new virus particles at its surface and then releases the particles so that they can infect other cells.

HIV-1 viruses encode their genetic information as molecules of ribonucleic acid (RNA). However, the host cell also makes many other RNA molecules that do not contain virus genes so there must be a mechanism in place to ensure that the new virus particles only contain viral RNA. An HIV-1 protein called Gag is responsible for assembling new virus particles and several Gag proteins come together on the cell membrane to form a honeycomb-like structure called the immature lattice. However, it is not clear how Gag is able to select the right RNA molecules.

To study how RNA is packaged into new HIV-1 particles, Carlson et al. used artificial versions of the cell membrane, viral RNA and the virus protein Gag to create a simple cell-free system. This system shows that all that is needed for viral RNA to be correctly packaged into new HIV-1 particles is for Gag to be attached to the cell membrane in such a way that the lattice forms correctly. Disturbing the immature lattice by altering the Gag proteins can result in a drastic loss of RNA selectivity. Further experiments show that other molecules in host cells called transfer RNAs enhance the ability of Gag to select the RNAs that encode virus genes.

Carlson et al.'s findings reveal a link between the formation of the Gag lattice and the packaging of virus genes into new virus particles. Drugs that inhibit this process could have the potential to be used as therapies against HIV-1. A future challenge will be to re-create the entire process of HIV-1 assembly in a cell-free system, which would make it easier to develop new drugs that target the process.

---

Single-virion fluorescence assays indicate that >90% of released HIV-1 particles have packaged genomic RNA (*Chen et al., 2009*), and packaging of a dimeric genome appears to be strongly preferred (*Sakuragi et al., 2003*; *Chen et al., 2009*; *Nikolaitchik et al., 2013*). The packaging of a single dimeric genome is independent of RNA mass, since genomes with altered lengths are always packaged as single dimers (*Nikolaitchik et al., 2013*). Genomic RNAs with sequence differences are efficiently packaged as heterodimers as long as they have identical DIS sequences (*Chen et al., 2009*). The ψ/SL3 stem loop region in the 5'UTR was initially identified as a sequence required for efficient packaging of genomic RNA into HIV-1 virions (*Lever et al., 1989*). Later work characterized a more extended core encapsidation signal (CES) which also includes the U5:AUG and PBS stems, the dimerization signal (DIS) and the splice donor (SD) region (*Heng et al., 2012*). Unpaired and weakly paired guanosines in several regions of the CES, including but not limited to ψ/SL3 were identified as important for packaging (*Abd El-Wahab et al., 2014*; *Keane et al., 2015*; *Kenyon et al., 2015*). Structures of NC complexed with ψ/SL3 (*De Guzman et al., 1998*) and other smaller fragments from the 5'UTR (*Lu et al., 2011b*) have provided atomic details of how the two zinc knuckles and adjacent motifs of NC interact with RNA. A crystal structure of the DIS:DIS dimerization complex (*Ennifar et al., 2001*) along with an NMR structure of the entire CES (*Keane et al., 2015*) put these structures in the context of a larger RNA. Thus, we now have a detailed understanding of RNA structural determinants for packaging at the level of individual 5'UTR dimers.

A single released HIV-1 virus contains one dimeric genomic RNA (*Chen et al., 2009*) and 2400 ± 700 Gag molecules (*Carlson et al., 2008*), indicating that the selective genome packaging may depend on more than just the binary NC-CES interaction. Indeed, cross-linking-immunoprecipitation sequencing (CLIP-seq) revealed a strikingly different RNA preference of cytosolic and membrane-bound Gag (*Kutluay et al., 2014*). Cytosolic Gag binds tRNA through its otherwise membrane-binding matrix (MA) domain, and associates mainly with cellular RNAs through its NC domain. The minor fraction of cytosolic Gag which binds viral RNA has a strong preference for the 5'UTR but also binds to the Rev-responsive element (RRE), whose main function is thought to be in nuclear export

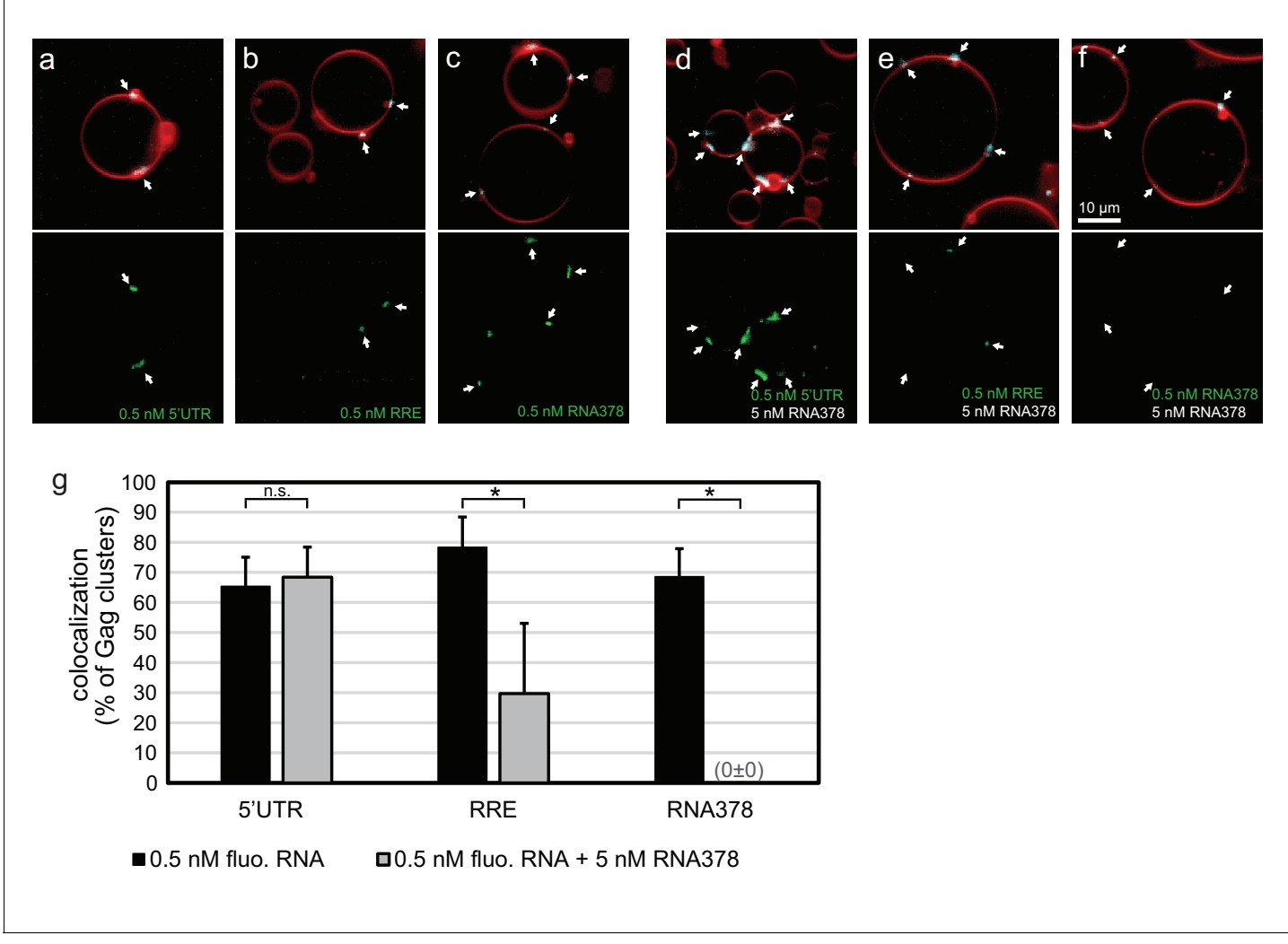

**Figure 1.** In vitro reconstitution of selective RNA packaging by HIV-1 Gag. (a) Confocal fluorescence micrograph of GUVs containing 5% PI(4,5)P$_2$. 100 nM HIV-1 Gag-ATTO594 and 0.5 nM HIV-1 5'UTR-Alexa488 were premixed and added to the exterior of the GUVs which were imaged 10 min later. Upper panel, membrane in red and Gag in white/cyan. Lower panel, RNA. (b–c) As (a) but with fluorescent HIV-1 RRE and a 378 nt control ssRNA (RNA378), respectively. (d–f) As (a–c) but with 5 nM non-fluorescent RNA378 added together with the fluorescent RNAs. (a–f) White arrows mark Gag clusters on GUVs, and their position on the corresponding RNA images. Scale bar for all images, 10 µm. (g) Quantitation of fluorescent RNA binding to Gag clusters. Gag clusters on membranes were identified in 10 z-stacks each containing ~10 GUVs, using an unsupervised script which calculated the average RNA fluorescence within the clusters. Gag clusters with an average RNA fluorescence >2.0 times that of the surrounding membrane were counted as positive for colocalization. All measurements were conducted on the same batch of GUVs and error bars indicate standard deviation between three repeats on separate GUV preparations. n.s./*, not significant, and significant, respectively, at p<0.05 level by Student's t-test.

The following figure supplement is available for figure 1:

**Figure supplement 1.** RNA recruitment by clusters of Gag with altered NC domain.

(**Kuzembayeva et al., 2014**). On the other hand, Gag in the membrane fraction has lost its tRNA association and is strongly enriched for viral RNA, interacting with equal probability with all parts of the genomic RNA. Existing structural and biochemical data are not sufficient to explain how a pool of cytosolic Gag, binding mainly cellular RNA, assembles into virus particles which package dimeric viral genomes with 90% probability.

Here, we have reconstituted the selective genome packaging of HIV-1 in a minimal system using giant unilamellar vesicles (GUVs), recombinant myristoylated HIV-1 Gag and purified RNAs. Gag assembling on GUV membranes selectively packages the HIV-1 5'UTR at subnanomolar

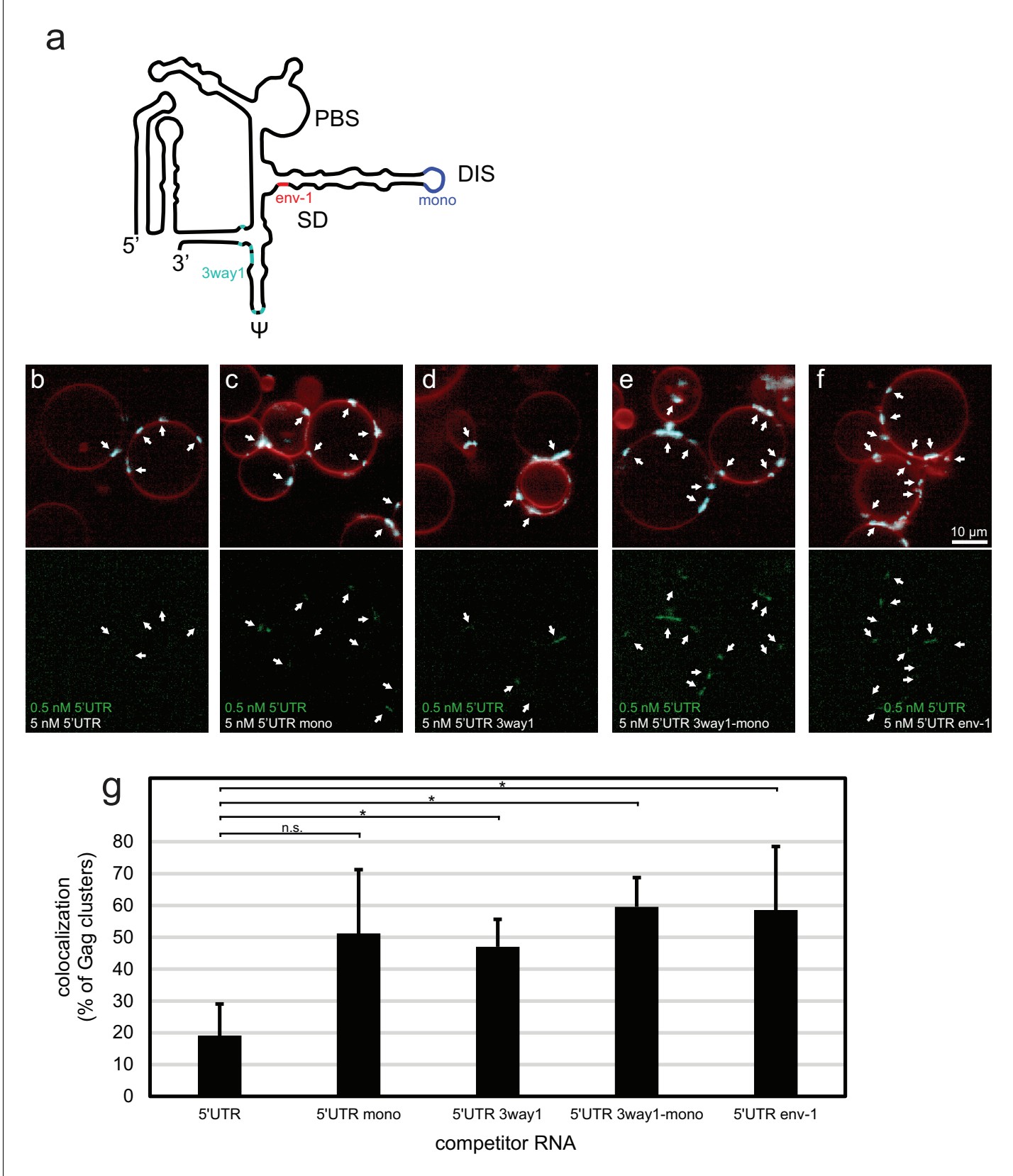

**Figure 2.** Effect of 5'UTR alterations on packaging by HIV-1 Gag. (a) Schematic of the HIV-1 5'UTR indicating introduced mutations. Blue indicates the position of the GCGCGC to GAGA mutation in the dimerization motif creating the monomerized construct '5'UTR mono'. Red indicates the position of
*Figure 2 continued on next page*

*Figure 2 continued*

the splice donor site G290. Replacement of nucleotides after G290 with the sequence from a spliced RNA created the construct '5'UTR env-1'. Cyan indicates the position of packaging-critical unpaired and weakly paired guanines mutated to adenines in the construct '5'UTR 3way1'. PBS, primer binding site. DIS, dimerization signal. SD, splice donor site. ψ, psi stem loop. The schematic shows one monomer of the dimerizing 5'UTR. (**b–f**) Confocal micrographs of GUVs, 10 min after adding 100 nM Gag-ATTO594, 0.5 nM 5'UTR -Alexa488, and 5 nM non-fluorescent competitor RNA as indicated. Upper panels, membrane in red and Gag in white/cyan. Lower panel, RNA. White arrows mark Gag clusters on GUVs, and their position on the corresponding RNA images. Scale bar for all images, 10 μm. (**g**) Quantitation of fluorescent RNA binding to Gag clusters, performed as for *Figure 1g* but using a lower threshold of 1.25 for counting Gag clusters as positive for fluorescent RNA. n.s./*, not significant, and significant, respectively, at $p < 0.05$ level by Student's t-test.

concentrations in the presence of excess competitor RNA. We find that the RNA selectivity of Gag is abolished by mutations that subtly affect its ability to cluster on a membrane. Using SHAPE analysis, we find that membrane recruitment of the 5'UTR by Gag does not lead to drastic changes in its secondary structure, and we further demonstrate that the association of soluble Gag with tRNA confers additional RNA packaging selectivity.

## Results

### In vitro reconstitution of selective RNA packaging by membrane-bound HIV-1 Gag

We previously reported an in vitro reconstitution of HIV-1 assembly using recombinant myristoylated Gag protein and PI(4,5)P$_2$-containing giant unilamellar vesicles (GUVs) (*Carlson and Hurley, 2012*). We showed that GUV-bound Gag clusters mimic the behavior of cellular HIV-1 budding sites in their lipid requirements for assembly, nucleic acid- and ESCRT (endosomal sorting complex required for transport) protein binding. We reasoned that this system could also be used to study the RNA selection of an assembling HIV-1 virus. The 5' untranslated region (5'UTR) and the Rev-responsive element (RRE) of the HIV-1 genome (nucleotides 1–345 and 7259–7612, of the genomic RNA of HIV-1 strain NL4-3 and ARV-2/SF2, respectively), along with a nominally unstructured 378 nt single-stranded control RNA (RNA378), were produced by in vitro transcription. The 1–345 construct was previously shown to dimerize even in the absence of Gag (*Heng et al., 2012*). RNAs were labeled with the fluorophore Alexa488 on random guanosines, at measured average labeling efficiencies of ~1.1–1.2 fluorophores per RNA molecule. 100 nM Gag-ATTO594 was premixed with 0.5 nM fluorescent RNA, and added to GUVs which were imaged by confocal microscopy starting 10 min after addition of protein and RNA. Due to the reported Mg$^{2+}$-dependence of the 5'UTR dimerization (*Wilkinson et al., 2008*), 2 mM MgCl$_2$ was included in the final buffer. The addition of Mg$^{2+}$ to the imaging buffer made the GUV morphology more variable, but this increased morphological heterogeneity did not hamper the experiments nor the image quantification.

In order to quantitate the RNA occupancy of assemblies, ten confocal z-stacks were recorded at random positions on each sample, with each data set containing an average of 50–100 GUVs. Strong RNA fluorescence was consistently observed at Gag clusters on GUV membranes, regardless of which RNA was used (*Figure 1a–c,g*). This RNA recruitment was dependent on the NC domain of Gag since neither a ΔNC Gag nor Gag treated with the Zn-finger disrupting drug AT-2 (*Rossio et al., 1998*) was able to recruit fluorescent RNA to GUV membranes (*Figure 1—figure supplement 1*). To study the RNA selectivity of assembling Gag in the reconstituted system, we included a ten fold excess (5 nM) of non-fluorescent RNA378 when premixing RNA and Gag. After assembly on GUVs, the fluorescent 5'UTR was still visible at most Gag clusters in the presence of ten fold excess competitor RNA, whereas RRE was present at fewer Gag clusters and at lower fluorescence intensities, and fluorescent RNA378 was virtually absent (*Figure 1d–f,g*). Image analysis based on automatic segmentation of Gag clusters and measurement of their average RNA fluorescence confirmed the visual impressions (*Figure 1g*).

To further characterize the sensitivity of the in vitro system, we created a set of altered 5'UTR constructs (*Figure 2a*). 5'UTR 3way1 has several packaging-critical guanines around the three-way junction 1 in the CES mutated to adenines (G116A, G318A, G320A, G328A, G329A, G331A, G333A). This set of mutations was previously found to reduce packaging in a cellular packaging

assay (*Keane et al., 2015*). 5'UTR mono has the GCGCGC sequence in DIS replaced by GAGA, creating a monomeric 5'UTR. 5'UTR 3way1-mono combines the 3way1 and mono mutations. Lastly, 5'UTR env-1 represents the first 345 nt of a spliced viral RNA, with the nucleotides after splice donor site G290 replaced by the nucleotides after splice acceptor site A5 (*Stoltzfus, 2009*). In a competition assay similar to the one reported in *Figure 1d*, 100 nM Gag-ATTO594 was premixed with 0.5 nM 5'UTR-Alexa488 and 5 nM of either non-fluorescent 5'UTR construct, and added to GUVs (*Figure 2b–f*). In the presence of 5 nM 5'UTR very little fluorescent 5'UTR could be seen at Gag clusters on GUVs (*Figure 2b,g*). However, in the presence of either of the altered constructs, more fluorescent 5'UTR was visible at Gag clusters (*Figure 2c–f,g*). Thus, the in vitro packaging assay is able to detect the lower affinity of assembling Gag for monomeric, 3way1-mutated, and spliced 5'UTRs.

In summary, a minimal system consisting of purified RNAs, Gag, and PI(4,5)P$_2$-containing GUVs recapitulates the selective RNA packaging of an assembling HIV-1 particle at RNA concentrations far below reported solution affinities of NC for 5'UTR. The 5'UTR is strongly selected for in a competition assay, whereas the RRE is only partially selected for after addition of competitor. 5'UTRs that are either monomerized, have packaging-critical guanines mutated to adenines, or represent a spliced viral RNA are less favored in the competition assay. Given the low concentrations at which this selectivity is observed, below the reported 17 nM affinity constant for 5'UTR-NC association (*Heng et al., 2012*), we turned our attention to potential avidity effects due to Gag multimerization.

## RNA selectivity depends upon intact CA domain lattice contacts

We next sought to determine whether there was a connection between RNA selectivity by Gag, and its assembly into the membrane-bound hexagonal 'immature' lattice found in viral budding sites and released immature virions (*Ganser-Pornillos et al., 2012*; *Sundquist and Kräusslich, 2012*; *Freed, 2015*) (*Figure 3a*). To this end, we produced Gag proteins mutated at symmetry contacts in its lattice-forming CA domain. The W316D,M317D mutation ("Gag two-fold") alters a motif at the two fold symmetry contact in the CA C-terminal domain (CA-CTD), which is described in the literature to be important for virus assembly (*Gamble et al., 1997*; *von Schwedler et al., 2003*) (*Figure 3a–b*). When this protein was added to GUVs at 100 nM, it bound in a strikingly different way from wild-type Gag, often resulting in a diffuse Gag fluorescence covering the entire GUV membrane (*Figure 3f,g*). For comparison, wild type Gag essentially only bound to GUVs in clusters even in the absence of RNA (*Figure 3e,g*). Based on the structure of the immature CA lattice in intact HIV-1 virions (*Schur et al., 2015*), we further produced a triple mutant ("Gag three-fold": P170E, A174E, E177R) designed to alter a three-helix interaction at the three fold interface in CA-NTD (*Figure 3a,c*). This protein bound GUVs similarly to wild-type Gag, albeit with a moderate increase in the diffuse binding first observed for Gag two-fold (*Figure 3g*). A Gag protein mutated at the six fold interface in CA-CTD (*Figure 3a,d*) with the single-residue replacement K290A ("Gag six-fold") reported to affect virus assembly (*von Schwedler et al., 2003*; *Robinson et al., 2014*), also showed an increase in diffuse Gag binding over wild-type Gag (*Figure 3g*). Combining the two-, three-, and six-fold mutations in one protein ("Gag 2-,3-,6-fold") resulted in an assembly phenotype similar to that of Gag two-fold (*Figure 3g*). Taken together, mutations in the CA-CTD of either the W316,M317 motif at the two-fold symmetry contact, or of K290 at the six-fold symmetry contact, both described as leading to virus release defects (*von Schwedler et al., 2003*), affect the ability of Gag to cluster on membranes.

Having characterized the membrane binding of the mutant Gag proteins, we proceeded to study their RNA binding and selectivity. We first compared the membrane binding of Gag proteins in the presence of RNA to Gag proteins alone as characterized above. We observed that the presence of RNA increased the clustering of all Gag proteins, and increased the fraction of GUVs which had bound Gag (*Figure 4a–j*, *Figure 4—figure supplement 1*). First, the capacity of the Gag proteins to recruit fluorescent RNA in the absence of competitor was studied. Each Gag construct was premixed with 0.5 nM fluorescent 5'UTR and added to GUVs. Recruitment of fluorescent 5'UTR to the GUV membrane was observed for all Gag constructs (*Figure 4a–e*), with a quantitation showing a slight decrease in recruitment for the Gag two-fold and Gag two-, three- , six-fold mutants (*Figure 4k*). However, when a ten fold excess of non-fluorescent competitor RNA (5 nM RNA378) was added, the Gag proteins behaved differently from each other. Wild type Gag retained as much 5'UTR fluorescence in the presence of competitor RNA as in its absence (*Figure 4f,k*) and the Gag three-fold mutant showed only a minor decrease (*Figure 4h,k*). On the other hand, GUV-bound clusters of the

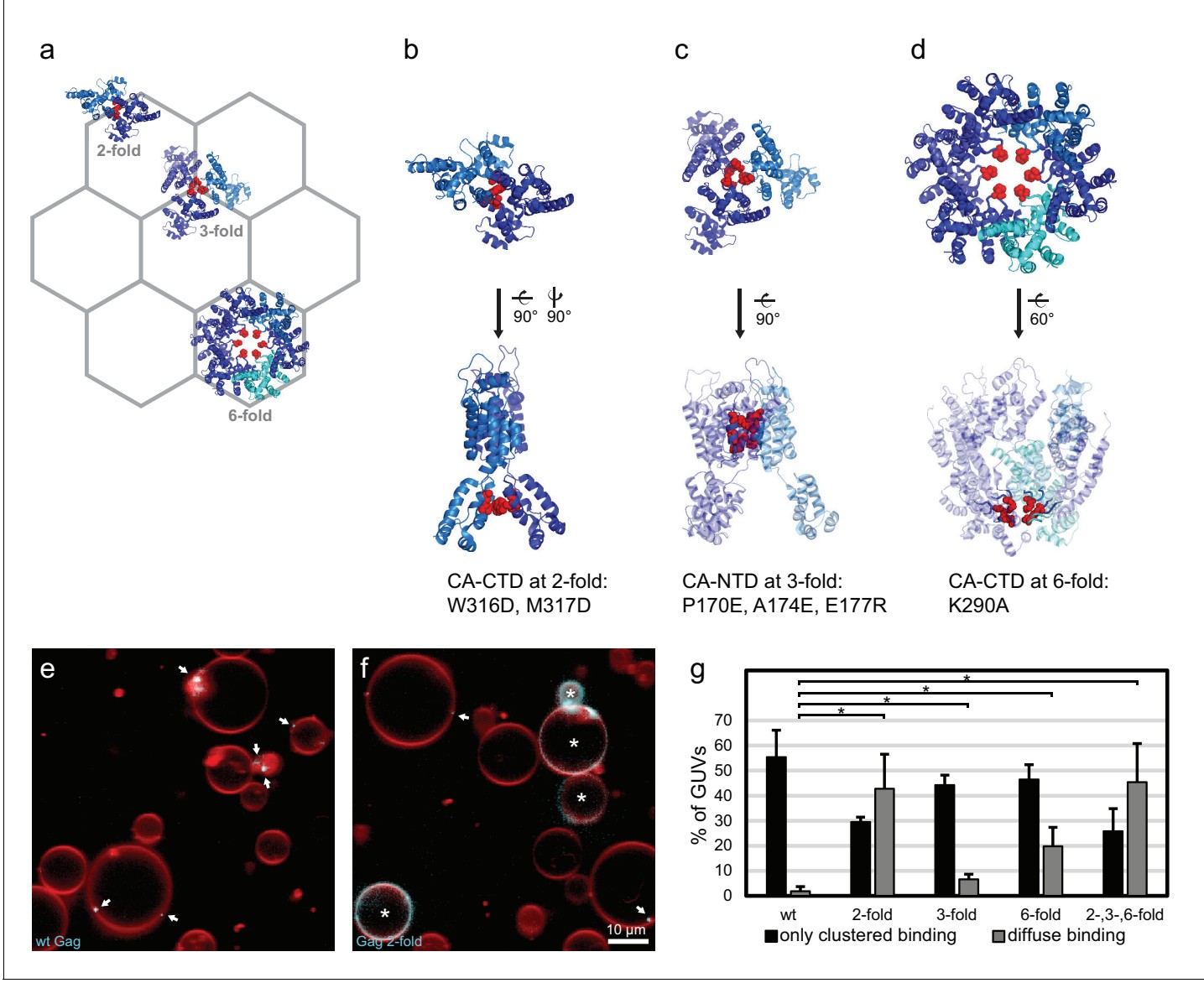

**Figure 3.** CA domain mutants affect Gag clustering on membranes. (a) Schematic of a hexagonal lattice showing the Gag CA domain at the two-, three- and six fold symmetry contacts, as arranged in the immature Gag lattice in HIV-1 assembly sites and immature virions (*Schur et al., 2015*) (PDB entry 4USN used in panel a–d). (b) Model of the CA domain at the two fold, with mutated residues in "Gag two-fold" marked in red. (c–d) As (b), for Gag mutated at the three- and six-fold symmetry contacts, respectively. (e) Confocal micrograph of GUVs, 10 min after adding 100 nM Gag-ATTO594. Membrane in red and Gag in white/cyan. (f) As (e), using Gag-ATTO594 two-fold. GUVs with diffuse Gag binding are denoted with white asterisks. (e–f) White arrows mark Gag clusters on GUVs. Scale bar, 10 μm. (g) Type of Gag binding to GUVs. For each protein, GUVs were counted in 10 confocal z-stacks, and classified according to having no Gag fluorescence, only clustered Gag fluorescence (black), or a diffuse Gag fluorescence covering the entire membrane (with or without additional brighter clusters, gray). All measurements were conducted on the same preparation of GUVs, with error bars indicating the standard deviation between three repeats conducted on separate GUV preparations. *significant at p<0.05 level by Student's t-test.

Gag two-fold, Gag six-fold and Gag two-, three-, six-fold mutants lost most of their 5'UTR fluorescence in the presence of competitor RNA (*Figure 4g,i–k*). Thus, RNA selectivity in this assay closely correlates with clustered binding of Gag to GUVs in the absence of RNA (compare *Figures 3g* and *4k*). In summary, mutations which cause subtle changes in the ability of Gag to multimerize on a membrane drastically decrease the RNA selectivity of assembled Gag clusters, without greatly affecting their ability to bind RNA non-specifically.

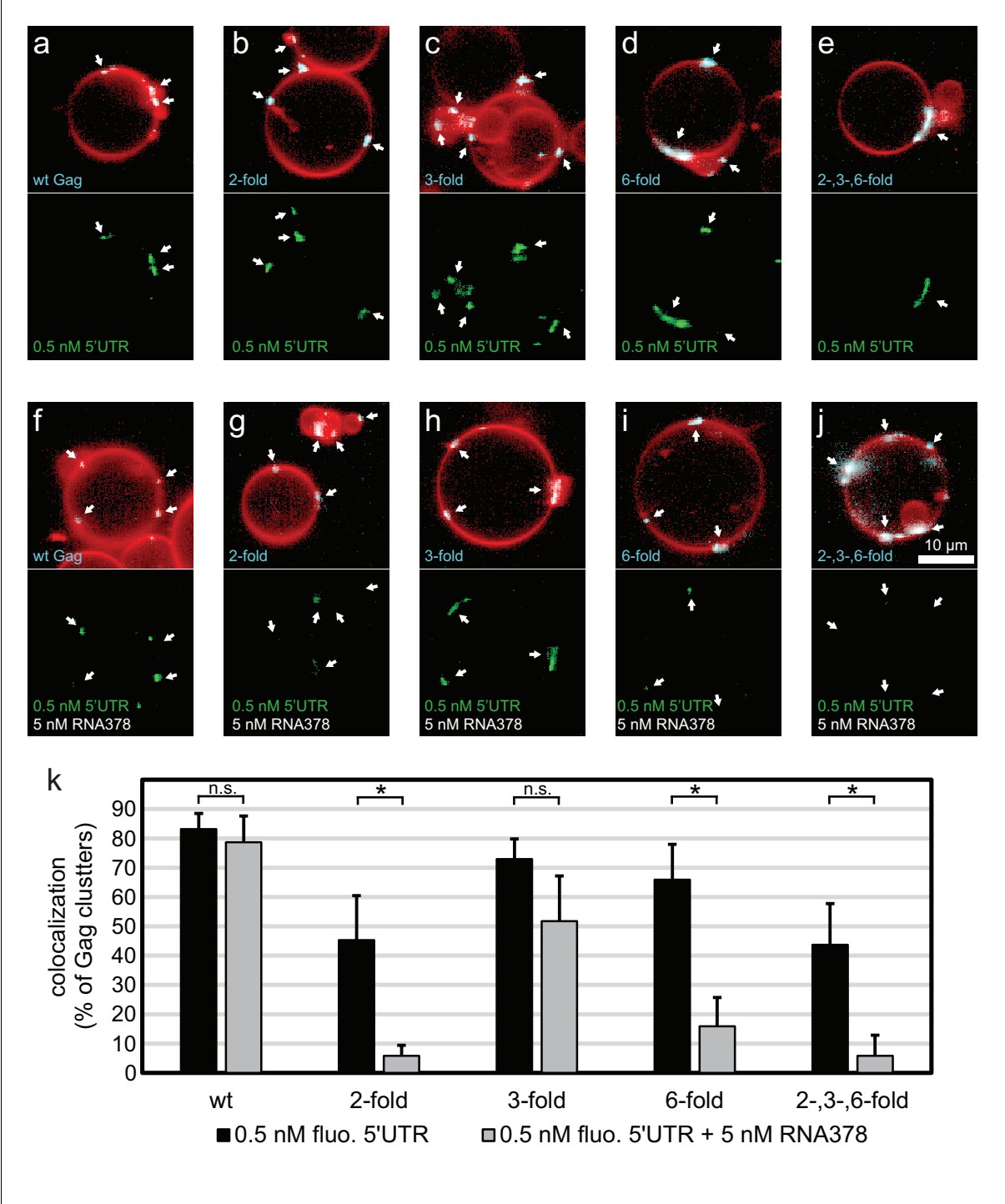

**Figure 4.** RNA selectivity of CA domain mutants. (a) Confocal micrograph of GUV, 10 min after adding 100 nM Gag-ATTO594 and 0.5 nM 5'UTR - Alexa488. Upper panel, membrane in red and Gag in white/cyan. Lower panel, RNA. (b–e) As (a) using Gag two-fold, Gag three-fold, Gag six-fold and Gag two-, three-, six-fold, respectively. (f–j) As (a–j) with the addition of 5 nM non-fluorescent RNA378. (a–j) White arrows mark Gag clusters on GUVs, and their position on the corresponding RNA images. Scale bar for all images, 10 μm. (k) Quantitation of RNA binding to Gag clusters, performed as for *Figure 1g*. n.s./*, not significant, and significant, respectively, at p<0.05 level by Student's t-test.

The following figure supplement is available for figure 4:

**Figure supplement 1.** Type of Gag binding to GUVs in the presence of RNA.

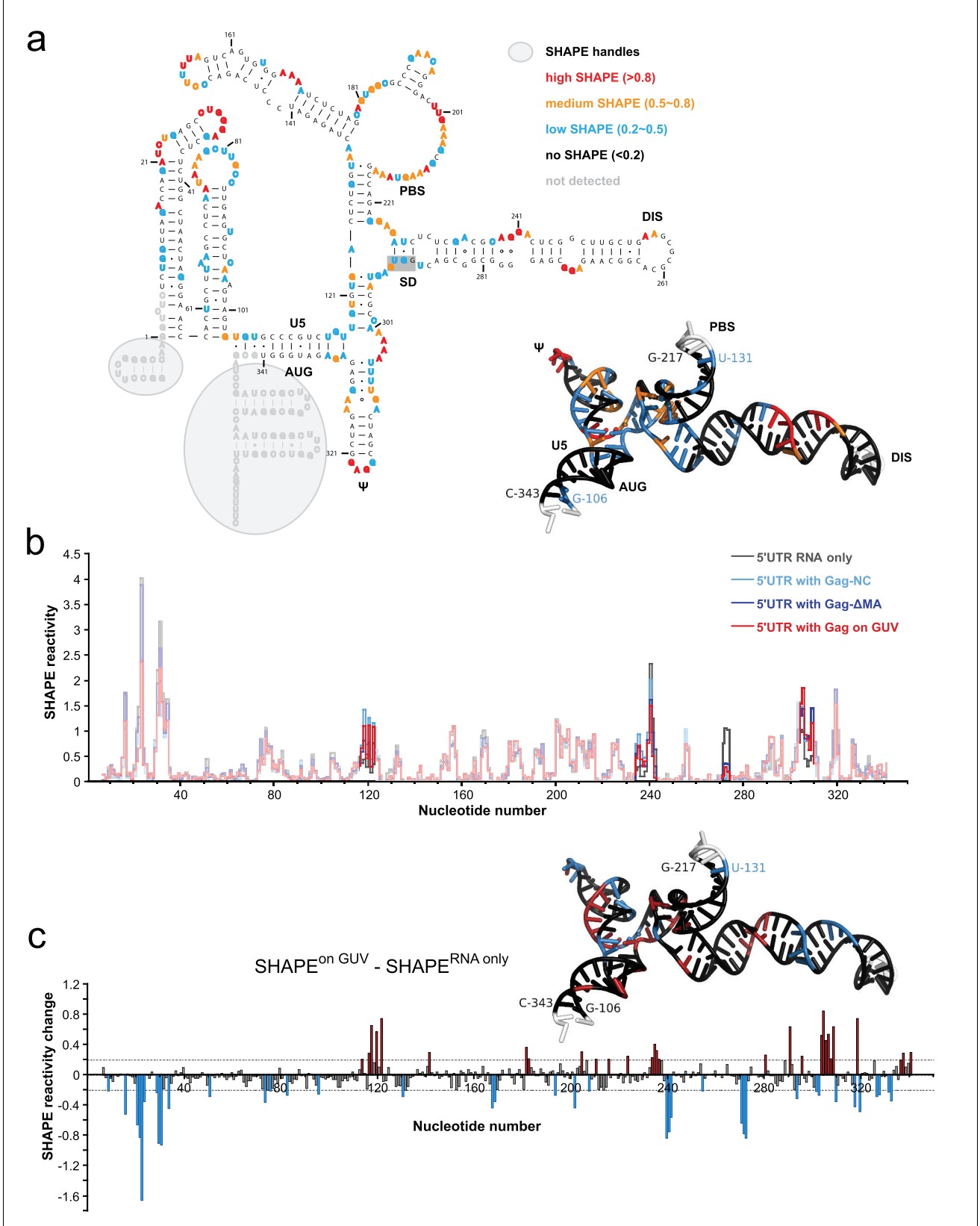

**Figure 5.** Probing Gag-induced structural changes on the HIV 5'UTR using SHAPE. (**a**) Secondary structure of the HIV 5'UTR RNA with SHAPE reactivities labeled. The secondary structure is based on the reported NMR structure (**Keane et al., 2015**) (PDB 2N1Q). Nucleotides exhibiting high,

*Figure 5 continued on next page*

*Figure 5 continued*

medium, low, or no SHAPE reactivities are labeled in red, orange, cyan, and black, respectively. SHAPE handles are labeled by gray shadows. The 3D structure colored based on the SHAPE profile using the same color scheme is shown as an insert. U5, U5 region of long terminal repeat. PBS, primer binding site. DIS, dimerization signal. SD, splice donor site. ψ, psi stem loop. AUG, start codon of gag gene. The nucleotides of SD that would be in a loop in the alternative conformation (289–292) are underlined with a gray box. The schematic shows one monomer of the dimerizing 5'UTR. (b) SHAPE profiles of the 5'UTR either alone (in gray) or in complex with the following Gag variants: the NC domain of Gag (in cyan), ΔMA-Gag (in blue), and Gag assembled on GUVs (in red). The regions not showing significant SHAPE changes are masked. (c) SHAPE changes between the 5'UTR alone and the 5'UTR bound by the Gag on GUV. Nucleotides exhibiting reduced or increased SHAPE value upon complex formation are labeled in cyan and dark red, respectively. Corresponding nucleotide are also labeled on the NMR 3D structure model using the same color scheme.

The following figure supplements are available for figure 5:

**Figure supplement 1.** SHAPE changes for protein addition in solution and a longer 5'UTR RNA.

**Figure supplement 2.** Gag binding to GUVs under conditions of the SHAPE analysis.

## SHAPE-based mapping of interactions between the HIV-1 5'UTR and Gag in solution and on membranes

To further understand how membrane association of Gag affects its interactions with the HIV 5'UTR, we performed selective 2'-hydroxyl acylation analyzed by primer extension (SHAPE) analysis (*Merino et al., 2005*) on the 5'UTR and determined its SHAPE profiles in the presence and absence of Gag constructs either in solution or in association with GUVs. First, we examined the SHAPE profile of the 5'UTR RNA in solution by itself. The RNA-only SHAPE profile is highly consistent with the recently reported solution structure of the HIV-1 RNA packaging signal (*Keane et al., 2015*), with the tandem three-way junctions exhibiting modest SHAPE reactivities, reflecting the loosely base-paired nature of this region. On the other hand, well based-paired stem regions in the DIS, PBS, U5: AUG, and ψ/SL3 sequences exhibit low or no SHAPE reactivities (*Figure 5a*). The dimer-promoting DIS loop (G257-C262) had no SHAPE reactivity. This is not per se an unequivocal indication of the 5' UTR being dimeric since DIS has been reported to partially base-pair with U5 in the monomeric conformation of the 5'UTR (*Lu et al., 2011a*). However, the SHAPE reactivity of the SD region G289-G292 (highlighted with a gray shaded box in *Figure 5a*) is consistent with the reported 3D structure of the dimeric conformation (*Keane et al., 2015*), not with the completely unprotected loop proposed for the monomeric 5'UTR (*Lu et al., 2011a*). A longer 356 nt 5'UTR able to assume both monomer- and a dimer-favoring conformations (*Heng et al., 2012*) had a similar SHAPE reactivity to the 345 nt 5'UTR (*Figure 5—figure supplement 1b*), indicating that it is present mainly as a dimer under the conditions of the study. We next compared the SHAPE profiles of isolated RNA, and RNA in complexes mimicking steps in the genome packaging. Isolated NC domain was selected to represent the initial interaction of a single Gag molecule with the 5'UTR, ΔMA(1–119)-Gag for a potential soluble Gag-RNA complex involving CA-mediated Gag-Gag interactions, and Gag clusters on GUVs as a model of viral budding sites on the plasma membrane. In all measurements, protein was kept in excess (solution protein:RNA ratio 8:1, GUV protein:RNA ratio 20:1). For a SHAPE analysis on GUV-bound RNA, reactions were set up as for confocal imaging, and GUVs were found to be intact during the treatment (*Figure 5—figure supplement 2*). The amount of free RNA in the GUV supernatant was typically 10–20% of the total RNA, dependent on the presence of both GUVs and Gag. The SHAPE profiles for 5'UTR were very similar in the presence of NC, ΔMA-Gag, and when bound to Gag clusters on GUVs (*Figure 5b*, *Figure 5—figure supplement 1*). Upon Gag binding, major SHAPE changes were observed in the following clusters: U117-U122, G234-A242, A271-G273, and A302-G309 (*Figure 5b*). Among them, the two clusters exhibiting decreased SHAPE reactivity (G234-C242 and A271-G273) surround the two guanine-rich bulges on the DIS stem. This Gag-mediated decrease of RNA flexibility is consistent with, but does not necessarily reflect, direct Gag-RNA contacts (*Kenyon et al., 2015*). These clusters show a more pronounced decrease in SHAPE reactivity than the ψ loop G317-G320. The other two clusters showing major SHAPE reactivity changes, U117-U122 and A302-G309, are located around the tandem three-way junctions. Both regions exhibit higher SHAPE reactivities upon Gag binding, indicating that protein interactions lead to rearrangement of the local structure that results in higher RNA flexibility. However, GUV recruitment of

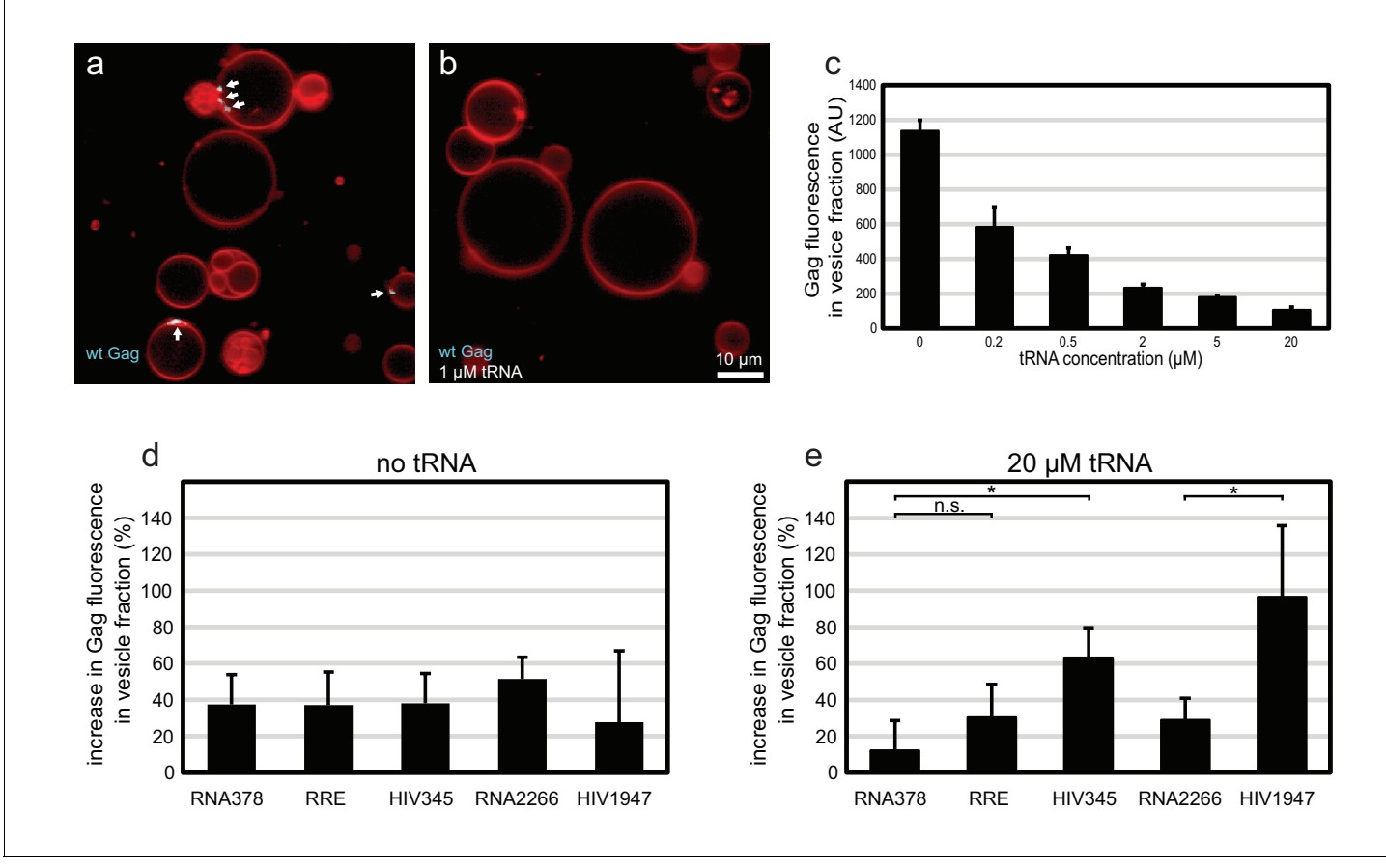

**Figure 6.** Effect of tRNA and viral RNA on Gag membrane association. (**a**) Confocal micrograph of GUVs, 10 min after adding 100 nM Gag-ATTO594. (**b**) Confocal micrograph of GUVs, 10 min after adding 100 nM Gag-ATTO594, premixed with 1 μM tRNA. (**a–b**) Membrane in red and Gag in white/ cyan. White arrows mark Gag clusters on GUVs. Scale bar, 10 μm. (**c**) Amount of Gag fluorescence in the vesicle fraction after a vesicle flotation assay with 100 nM Gag-ATTO488, tRNA at 0–20 μm, and 0.4 mg/mg 1.0 μm LUVs. (**d**) Amount of Gag fluorescence in the vesicle fraction after a vesicle flotation assay with 100 nM Gag-ATTO488 and RNAs at 5 nM. (**e**) Amount of Gag fluorescence in the vesicle fraction after a vesicle flotation assay with 100 nM Gag-ATTO488, 20 μm tRNA, and other RNAs at 5 nM. (**c–e**) All measurements shown in the same panel were conducted on the same LUV preparation, with error bars indicating the standard deviation between three repeats conducted on separate LUV preparations. n.s./*, not significant, and significant, respectively, at p<0.05 level by Student's t-test.

the complex did not trigger additional SHAPE changes (*Figure 5c*, *Figure 5—figure supplement 1*), indicating that the overall structure of the 5'UTR is not further altered between the states of initial Gag binding in solution and recruitment to assembled Gag on a membrane. Noteworthy, the SHAPE reactivity of the SD region G289-G292 is similar under all conditions studied (*Figure 5a,c*, *Figure 5— figure supplement 1*), indicating that the 5'UTR is maintained in the dimeric conformation over the assembly process. Taken together, we see no evidence for large changes in secondary structures in the 5'UTR when interacting with either Gag in solution or assembled Gag on a membrane, and it appears dimeric under the conditions studied.

## The effects of tRNA and viral RNAs on Gag membrane association

The membrane-binding matrix (MA) domain of cytosolic Gag has been shown to bind primarily tRNA in cells (*Kutluay et al., 2014*). Several reports have shown that RNA binding to the basic patch on the MA domain of HIV-1 Gag can compete with its ability to bind acidic phospholipids such as PI (4,5)$P_2$ (*Alfadhli et al., 2009*, *2011*; *Chukkapalli et al., 2010*, *2013*; *Dick et al., 2013*; *Dick and Vogt, 2014*). We wanted to investigate whether Gag-tRNA interactions also play a role in selective RNA packaging. Premixing 100 nM Gag-ATTO594 with 1 μM yeast tRNA before addition to GUVs led to a virtually complete block in Gag assembly on GUVs (*Figure 6a–b*). Since the amount of Gag

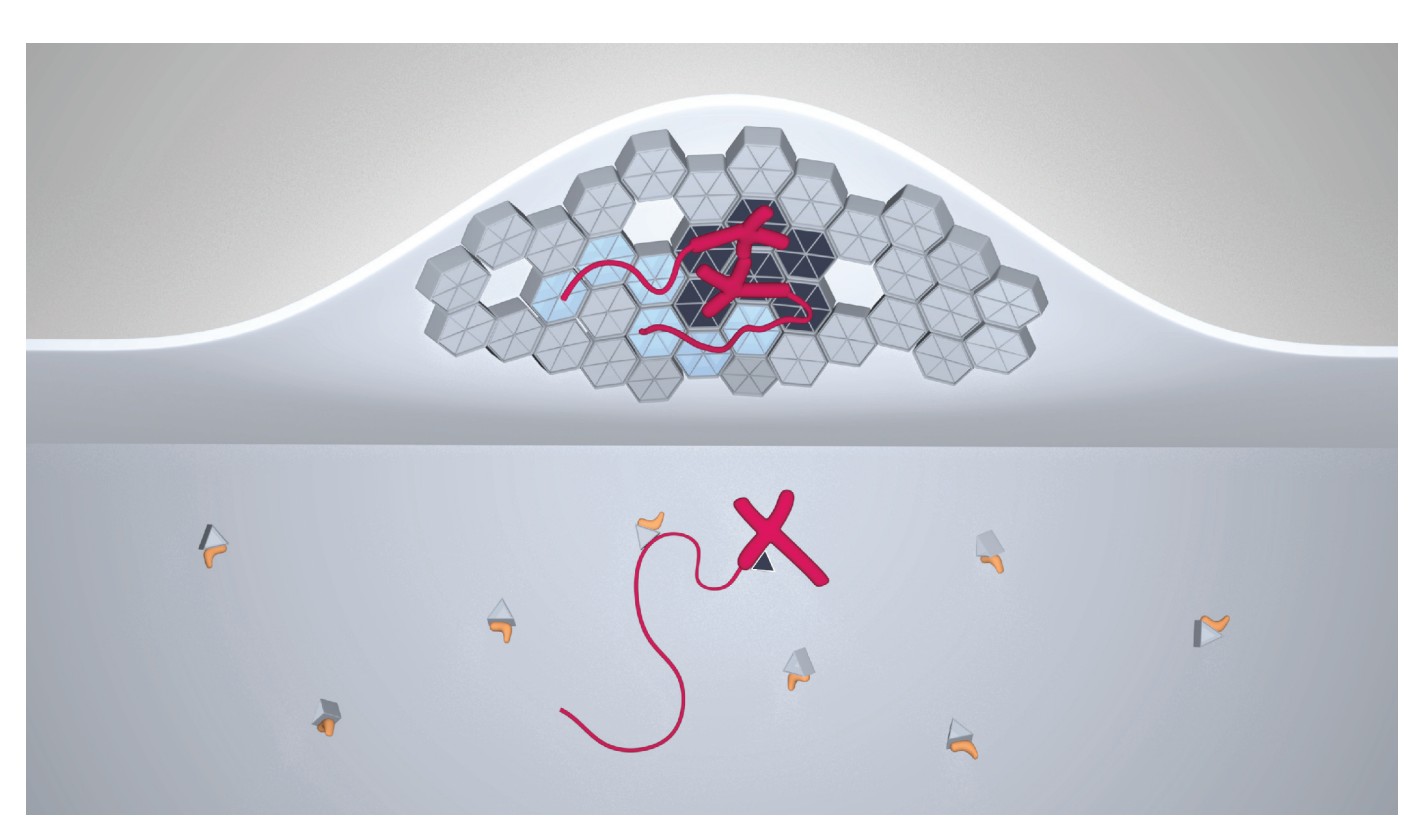

**Figure 7.** A model for selective HIV-1 genome packaging. Cytosolic Gag (light gray triangles) is inhibited from membrane association by its association with tRNA (yellow). Gag (triangles and hexagons, colored dark gray where forming specific contacts with the packaging signal) binding viral genomic RNA (red; thick portions correspond to the 5'UTR, thin portions to the remainder, not shown to scale) has a higher probability to overcome this inhibition and associate with the plasma membrane. The selectivity of membrane-bound Gag for the viral genome necessitates CA-CA immature lattice contacts.

on individual GUVs is more variable than the relative RNA fluorescence at Gag clusters, a membrane flotation assay was devised to quantitate the amount of Gag assembled on membranes. 100 nM Gag-ATTO488 was mixed with 0–20 µM tRNA, and incubated for 10 min with 1.0 µm large unilamellar vesicles (LUVs) containing 5 mol% PI(4,5)P$_2$. Vesicles were subjected to flotation on a density step gradient and Gag fluorescence in the top fraction was measured. For the tRNA titration, a 50% inhibition of Gag membrane association was evident at 200 nM tRNA (*Figure 6c*). At 20 µM tRNA, the amount of Gag found in the vesicle fraction dropped to ~10% of the value seen in the absence of RNA (*Figure 6c*). Next, long viral or unspecific RNAs at 5 nM concentration were added to Gag with or without tRNA. In addition to the above studied 5'UTR, RRE, and RNA378, an RNA corresponding to the first 1947 nucleotides of the HIV-1 strain NL4-3 genomic RNA was studied (HIV1947), along with a single-stranded control RNA of similar length (RNA2266). In the absence of tRNA, all these RNAs slightly increased the amount of Gag associated with vesicles (*Figure 6d*). In the presence of 20 µM tRNA however, the 5'UTR and the longer HIV1947 RNA (which contains the 5'UTR sequence) enhanced membrane association by Gag more than the other RNAs (*Figure 6e*). RNA378 had no effect on Gag membrane association in the presence of 20 µM tRNA, whereas the effect of RRE or the longer RNA2266 was slight (*Figure 6e*). In summary, tRNA at micromolar concentrations potently inhibits membrane association of Gag. Under such conditions, Gag that encounters viral RNA has a higher probability of overcoming that inhibition and associating with membrane.

## Discussion

Here, we used biochemical reconstitution methods to show that a minimal system consisting of the HIV-1 Gag protein, the HIV-1 5'UTR, and PI(4,5)P$_2$-containing GUVs is capable of selective RNA packaging in the presence of an excess of non-specific RNA. We find a strong correlation between the ability of Gag to multimerize on a membrane and its selective packaging of viral RNA. Mutations in the lattice-forming CA domain that lessen Gag clustering on membranes (W316D,M317D and K290A, *Figure 3b,d,f,g*) drastically lowered its RNA selectivity (*Figure 4g,i,k*), without affecting the total capacity of Gag to recruit RNA to a membrane (*Figure 4b,d,k*). Mutations of these residues in the C-terminal part of CA (CA-CTD) have previously been described to lead to virus release defects related to aberrant virus assembly (*Gamble et al., 1997*; *von Schwedler et al., 2003*). In line with our findings, it has been reported that the amount of viral RNA coprecipitated with cytoplasmic Gag is reduced five fold when its CA-CTD is deleted, whereas the amount of cellular RNA coprecipitated remains the same (*Kutluay and Bieniasz, 2010*). On the other hand, in experiments which did not address selectivity, a ΔCA-CTD Gag was still able to tether genomic RNA to the plasma membrane in cells (*Jouvenet et al., 2009*; *Sardo et al., 2015*), which is consistent with our finding that the mutant Gag proteins can still recruit 5'UTR although being less selective in the presence of competitors. Mutations at the three-fold symmetry contact in CA-NTD did not alter Gag clustering on membranes (*Figure 3g*), concurring with observations that a ΔCA-NTD Gag still supports particle release (*Borsetti et al., 1998*; *Accola et al., 2000*).

Live-cell fluorescence microscopy has delivered important constraints on the stoichiometry and the location of the RNA selection event. Genomic RNA is present at the onset of HIV-1 assembly at the plasma membrane, at which point the number of Gag molecules was reported to be below the detection limit of ~12 molecules (*Jouvenet et al., 2009*). Cytosolic Gag does not form large oligomers (*Robinson et al., 2014*; *Hendrix et al., 2015*) and mainly associates with cellular RNAs (*Kutluay et al., 2014*). A small fraction of cytosolic Gag is found to be slowly diffusing and consist of oligomers of up to ~5 molecules (*Hendrix et al., 2015*). In a recent study, TIRF microscopy was used to study the monomer-dimer state of genomic HIV-1 RNAs (*Chen et al., 2016*). It was reported that the large majority of plasma membrane-proximal genomic HIV-1 RNAs are monomeric in the absence of HIV-1 Gag. In cells expressing Gag, monomeric genomic RNAs are seen to coalesce to dimers on the membrane, but were not observed to arrive to the membrane as dimers. That study and our results underscore the importance of membrane-localized Gag assembly events in HIV-1 genome selection and packaging.

The SHAPE analysis focused on the secondary structure of the 5'UTR RNA in situations representing different stages of packaging (*Figure 5a,b*). When comparing SHAPE reactivities of isolated 5'UTR, 5'UTR interacting with NC domain in solution, ΔMA-Gag in solution, or recruited to GUV membranes by clusters of full-length Gag, the changes in SHAPE reactivities were consistent with a conservation of secondary structure between all these states. Based on these results, there is no gross unfolding of the dimeric 5'UTR in the early steps of genomic RNA selection and packaging for HIV-1. This contrasts with reports that the packaging signal of the related retrovirus Murine Leukemia Virus (MLV) is partially unfolded in released immature virions (*Grohman et al., 2014*). The SHAPE reactivity pattern indicated that the 5'UTR was, as expected, dimeric in solution. This was observed both with the shorter 345 nt 5'UTR, which has been reported to be locked in a dimer-favoring conformation, and the longer 356 nt 5'UTR which is also capable of assuming a second conformation (*Lu et al., 2011a*; *Heng et al., 2012*) (*Figure 5—figure supplement 1*). In the light of a recent report that most genomic HIV-1 RNAs in cells are monomeric until interacting with Gag on a membrane (*Chen et al., 2016*), the isolated 5'UTR would thus be representative of the genome after engagement with Gag. It is possible that the RNA chaperone activity of Gag may be necessary to expose the 5'UTR and its DIS in the context of a full-length genome. The strongest decrease in SHAPE reactivity upon Gag interaction was found in the guanosine-rich bulges on the DIS stem (*Figure 5c*), consistent with a recent study using full-length Gag in solution (*Abd El-Wahab et al., 2014*). The increased SHAPE reactivity in central regions of the tandem three-way junction suggests that Gag engagement, while not leading to a drastic rearrangement of the 5'UTR secondary structure, may lead to alterations its 3D structure. Overall, our results show that RNA conformational changes upon binding to membrane-associated Gag are minor in magnitude and thus suggest that conformation rearrangements are likely to be a minor contributor to selective packaging.

Cytosolic RNA has been reported to regulate Gag membrane association, effectively increasing its selectivity for the plasma membrane lipid PI(4,5)P$_2$ (*Alfadhli et al., 2009*, *2011*; *Chukkapalli et al., 2010*, *2013*; *Dick et al., 2013*; *Dick and Vogt, 2014*). The RNA species that binds the Gag MA domain in cells was identified by deep sequencing as primarily consisting of tRNA (*Kutluay et al., 2014*). Our reconstituted system allowed us to disentangle the effect that different RNAs have on Gag membrane association (*Figure 6*). We show that Gag-tRNA interactions may also serve to increase Gag fidelity for viral RNA. In the absence of tRNA, any long RNA caused a slight increase in Gag membrane association. But in the presence of 20 μM tRNA, 5'UTR and a longer 1947 nt fragment of the HIV-1 genome increased Gag membrane association more than other RNAs. We hypothesize that a viral RNA with several high-affinity binding sites for Gag can recruit several Gag molecules to the membrane if only one of them dissociates from its membrane-occluding tRNA.

In summary, our study leads to a model for HIV-1 genome selection where the NC domain of Gag is necessary but not sufficient for the selective packaging of the HIV-1 genomic RNA (*Figure 7*). Lattice contacts conferring Gag ability to multimerize on a membrane are crucial for the selective packaging of viral RNA. The 5'UTR is recruited to membranes by multimerized Gag without any gross changes to its secondary structure. Moreover, the association of Gag with tRNA serves to inhibit genome-less virus assembly.

# Materials and methods

## Protein expression and purification

Full-length, N-terminally myristoylated HIV-1 Gag was purified as previously described (*Carlson and Hurley, 2012*). Gag was labelled with ATTO488-maleimide or ATTO594-maleimide (Sigma-Aldrich, St. Louis, MO) on an engineered cysteine (A120C) in the MA-CA linker region. The resulting fluorophore-labeled protein eluted as a monodisperse peak from the final size exclusion column. ΔNC Gag was cloned from the expression plasmid for full-length Gag by deleting the sequence corresponding to the two Zn fingers of NC (residues 393–448 of full-length Gag, with the initiator Met being residue 1), and expressed and purified analogously to full-length Gag. The nucleocapsid (NC) domain of Gag (residues 378–432 of full-length Gag) was expressed in *E. coli* as a C-terminal, TEV-protease cleavable, maltose-binding protein(MBP)-(His)$_6$ fusion protein and purified analogously to full-length Gag, using NiNTA- and Heparin affinity columns followed by TEV protease digestion and size exclusion chromatography on a Superdex 75 16/60 column (GE Healthcare, Uppsala, Sweden). ΔMA Gag (residues 120–500) was based on the same synthetic gene as used for full-length Gag, expressed in *E. coli* as a TEV-cleavable (His)$_6$-MBP- ΔMA Gag-MBP construct, and purified as full-length Gag.

## RNA generation and fluorophore labeling

Yeast tRNA was purchased from Invitrogen. All other RNAs were generated by in vitro transcription using T7 polymerase, and purified either by sequential LiCl and ethanol precipitation, or polyacrylamide gel electrophoresis. The 5'UTR and RRE RNAs consisted of nucleotides 1–345 and 7259–7612 of the NL4-3 (GenBank: AF324493.2) and ARV-2/SF2 (GenBank: K02007.1) unspliced RNAs, respectively (*Heng et al., 2012*; *Bai et al., 2014*). RNA378 and RNA2266 were transcribed from the first 378 nucleotides of a synthetic gene encoding human VPS25 (*Im and Hurley, 2008*), and the first 2266 nucleotides of the gene encoding *H. sapiens* AIP1/ALIX (GenBank: NM_013374.5), respectively. The integrity of the RNAs was monitored by denaturing formaldehyde-agarose gel electrophoresis. Fluorophore labeling of RNA was carried out using ULYSIS Alexa Fluor 488 (Molecular Probes, Waltham, MA) at 90°C for 10 min in the presence of 3 mM MgCl$_2$ followed by rapid cooling to 4°C to refold the labeled RNAs. Labeled RNAs were buffer-exchanged using Bio-Spin 6 spin columns (BIO-RAD, Hercules, CA), and the final degree of labeling was measured using UV-VIS spectrometry.

## Vesicle preparations

Lipids were acquired from Avanti polar lipids (Alabaster, AL), except the fluorophore ATTO647-DOPE which was acquired from ATTO-TEC GmbH (Siegen, Germany). GUVs with a molar

composition of 69.9% palmitoyl-oleoyl-phosphatidylcholine (POPC), 25% cholesterol, 5% brain-extracted phosphatidylinositol-4,5-bisphosphate (PI(4,5)P$_2$) and 0.1% ATTO647-dioleoyl-phosphatidylethanolamine were electroformed as described previously (*Carlson and Hurley, 2012*). For LUVs, lipids with a molar composition of 70% POPC, 25% cholesterol, and 5% PI(4,5)P$_2$ were dried over night in a desiccator and resuspended in LUV buffer (20 mM Tris, 150 mM NaCl, pH 7.4). They were subjected to ten freeze-thaw cycles on liquid nitrogen, followed by eleven times extrusion through a Nuclepore Track-Etched Membrane with 1.0 μm pore size (Whatman, Little Chalfont, United Kingdom).

## Reconstitution on GUVs and confocal microscopy

GUVs electroformed in 600 mM sucrose were stored at room temperature and imaged within 24 hr. Protein and RNAs were preincubated for 5 min at room temperature in GUV dilution buffer (20 mM Tris, 300 mM NaCl, 4 mM MgCl$_2$, pH = 7.4) and then diluted 1:1 with GUVs in a Lab-Tek II chambered coverglass (Fisher Scientific, Waltham, MA). Ten minutes after mixing, the reaction was imaged using an LSM 5 LIVE confocal microscope (Carl Zeiss, Jena, Germany). The membrane fluorophore was imaged using the 635 nm laser and a 650 nm long-pass emission filter, Gag using the 532 nm laser and a 550–615 nm emission filter and RNA using the 488 nm laser and a 500–525 nm emission filter. Membrane, protein, and RNA were imaged using separate tracks, and cross-excitation was found to be minimal with the chosen imaging parameters. On each reaction, ten z-stacks (each consisting of ten slices with 1.0 μm spacing) were acquired at positions selected solely on the basis of containing GUVs, without observing the fluorescence channels prior to acquisition. All experiments shown in the same figure were done within 24 hr using the same GUV batch for comparability. Each such experiment series was repeated on three separate occasions with different GUV preparations.

## AT-2 treatment of Gag

To Gag-ATTO594, AT-2 (aldrithiol-2, Sigma-Aldrich) dissolved in DMSO was added to a final concentration of 100 μM, followed by a 15 min incubation at 37°C prior to imaging.

## Image analysis

Statistics on RNA recruitment to Gag clusters was performed as described previously (*Carlson and Hurley, 2012*). Briefly, a custom-written script for MATLAB (Mathworks, Natick, MA) used the confocal z-stacks to create a binary mask excluding non-membrane voxels, then identified continuous areas with high Gag fluorescence within the membrane mask. Such an area (Gag cluster) was considered positive for RNA recruitment if its average RNA fluorescence was >2.0 times higher than the average in the GUV membrane outside Gag puncta, except for *Figure 2* where a lower threshold of 1.25 was used. To quantitate the Gag binding type (*Figure 2g–h*), all GUVs with a diameter >5 μm were counted. The GUVs were inspected for Gag binding in the 532 nm fluorescence channel, and those GUVs with Gag clusters but areas completely devoid of detectable Gag fluorescence were counted as 'clustered only' The remaining GUVs either had no detectable Gag binding, or had an even fluorescence covering the entire membrane, sometimes with additional brighter areas. These GUVs were counted as 'diffuse binding'.

## SHAPE analysis of the interactions between the HIV-1 5'UTR and the Gag proteins

For SHAPE analysis, SHAPE handles were added to both ends of the 345-nt and 356-nt 5'UTR construct as previously reported (*Bai et al., 2014*). Purified RNA samples were annealed at 0.1 mg/ml in a buffer containing 50 mM HEPES-KOH pH 7.5, 200 mM KOAc, and 3 mM MgCl2 by heating at 75°C for 2 min and snap cooling on ice. Before SHAPE reactions, 9 μl of annealed 5'UTR RNA at 0.1 mg/ml was mixed with 1 μl of variants of Gag protein to achieve a final RNA:protein stoichiometry of ~1:8 (0.8 μM:7.2 μM). Matching protein buffer containing 20 mM Tris, 150 mM NaCl, 0.1 mM TCEP, pH = 7.4 was used as a control. The resulting mixtures were incubated for 15 min at room temperature. SHAPE probing was performed as previously reported (*Berry et al., 2011*) with 1-methyl-7-mitroisatoic anhydride (1M7) as the 2' hydroxyl-selective electrophile. For SHAPE analysis of GUV-bound RNA, reactions were set up in Lab-Tek II chambered coverglasses as for confocal

imaging to ensure identical mixing kinetics and GUV settling as during imaging. Gag and 5'UTR were premixed with GUV dilution buffer to 150 µl total volume and after 5 min at room temperature were added to 150 µl GUVs in imaging chambers for final concentrations of 100 nM Gag and 5 nM 5'UTR. 10 min after mixing, 200 µl supernatant was gently aspirated. 15 µl 1M7 in DMSO was briefly premixed with 35 µl GUV dilution buffer to avoid exposing GUVs to undiluted DMSO, added to the remaining 100 µl GUV solution in the chamber, and mixed by gentle stirring. Eight chambers were used for each reaction. The control reaction was performed by adding DMSO only. For the test of GUV integrity after this treatment (*Figure 5—figure supplement 2*), the control reaction was repeated with fluorescent Gag-ATTO594 and imaged by confocal microscopy as described above 5 min after DMSO addition. To measure the amount of free RNA in the GUV reactions, the supernatant which was removed prior to SHAPE reagent addition was centrifuged for 5 min in a tabletop centrifuge at 21,000 $\times$g to pellet any remaining vesicle-bound RNA. The RNA concentration in the supernatant of this sample was measured using a Qubit fluorimeter (ThermoFisher Scientific, Waltham, MA). Raw traces from fragment analysis was analyzed using ShapeFinder (*Vasa et al., 2008*).

## Vesicle flotation assay

Gag and RNA were mixed in a total volume of 60 µl LUV buffer containing 1.67 mM $MgCl_2$ and incubated for 5 min at room temperature. 40 µl 1.0 mg/ml freshly prepared LUVs were added and the samples were incubated for 10 min at 37°C. The reactions were mixed with OptiPrep solution (Sigma-Aldrich) and LUV buffer to 700 µl and 30% (w/v) iodixanol. This solution was placed at the bottom of ultracentrifuge tubes and overlaid with 900 µl 18% (w/v) iodixanol in LUV buffer and 700 µl LUV buffer. The gradients were subjected to ultracentrifugation in an Sw55.Ti rotor for 3 hr at 45,000 rpm at 4°C. At the end of the run the vesicles fractions were collected from the interface between the 0% and 18% iodixanol fraction, and the amount of Gag associated with the vesicles was measured in a GloMax-Multi microplate reader (Promega, Fitchburg, WI) using 490 nm excitation light and a 510–570 nm emission filter. All experiments shown in the same panel were conducted with the same preparation of LUVs and each experiment series was repeated on three separate occasions with different LUV preparations

## Acknowledgements

We thank Michael F Summers (University of Maryland, Baltimore County) and Johannes Schöneberg (UC Berkeley) for fruitful discussions and A Yeremenko for drawing *Figure 7*. This work was supported by the National Institutes of Health under award numbers R01AI112442 (JHH), P50GM082250 (JAD) and P50GM103297(MFS).

## Additional information

### Funding

| Funder | Grant reference number | Author |
|---|---|---|
| National Institute of Allergy and Infectious Diseases | R01AI112442 | James H Hurley |
| National Institute of General Medical Sciences | P50GM082250 | Jennifer A Doudna |
| National Institute of General Medical Sciences | P50GM103297 | Sarah C Keane |

The funders had no role in study design, data collection and interpretation, or the decision to submit the work for publication.

### Author contributions

L-AC, Conception and design, Acquisition of data, Analysis and interpretation of data, Drafting or revising the article, Contributed unpublished essential data or reagents; YB, Conception and design, Acquisition of data, Analysis and interpretation of data, Drafting or revising the article; SCK, Drafting

or revising the article, Contributed unpublished essential data or reagents; JAD, JHH, Conception and design, Analysis and interpretation of data, Drafting or revising the article

**Author ORCIDs**

James H Hurley, http://orcid.org/0000-0001-5054-5445

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
