## [Decision Letter]

Thank you for submitting your article "Reconstitution of selective HIV-1 RNA packaging in vitro by membrane-bound Gag assemblies" for consideration by *eLife*. Your article has been favorably evaluated by Wenhui Li as the Senior editor and three reviewers, one of whom, Wes Sundquist, is a member of our Board of Reviewing Editors.

The reviewers have discussed the reviews with one another and the Reviewing Editor has drafted this decision to help you prepare a revised submission.

General assessment:

The authors describe a reconstituted HIV RNA "packaging" system featuring specific interactions between the viral RNA and membrane-associated Gag polyprotein. Attractive features of the system include: 1) dependence upon the 5'UTR core packaging site, 2) a requirement for immature Gag assembly, and 3) tRNA inhibition of Gag-membrane interactions that is specifically overcome by RNAs that contain the HIV-1 5'UTR. In addition to describing the system, the authors point out that their study is the first to show that the structure of the dimeric RNA does not change significantly during the different steps of packaging, as evidenced by a relative lack of SHAPE reactivity changes. This report marks the next progression in the excellent studies from the Hurley lab in which they have reconstituted different stages of HIV-1 assembly in vitro.

In principle, these studies represent a significant advance, but several additional controls are necessary to ensure that the system is accurately mimicking viral RNA packaging as it occurs in the context of a cellular infection. Specifically, recent studies from other laboratories support a model in which HIV RNA packaging occurs through a stepwise pathway in which the viral RNA begins in a translatable conformation (Gag AUG codon exposed), converts into a dimerizing conformation (Gag AUG codon sequestered, DIS element exposed), interacts specifically with a small number of Gag molecules that bind guanosine clusters in the 5'UTR, and then interacts with a much larger number of Gag molecules along its entire length as the immature virion lattice assemblies at the membrane. Unspliced, dimeric RNA is selectively packaged against a background of cellular RNAs, spliced RNAs, and monomeric RNAs. Thus far, the in vitro system has been shown to recapitulate some, but not all of these features.

Substantive concerns to be addressed:

1) The authors argue that their work "will stand as the definitive in vitro study of HIV genome packaging". This is certainly a worthy goal, but to achieve it the authors need to do more to demonstrate that their system really does recapitulate all of the key aspects of specificity seen in the cellular context.

Specifically, the authors need to show:

A) Selective packaging of dimeric vs. mutant HIV RNAs (e.g., using DIS mutants that are shown experimentally not to dimerize and using other mutant(s) that have been shown to reduce packaging).

B) Selective packaging of unspliced RNAs (e.g., using mimics of spliced viral RNAs).

C) That NC mutants (or AT-2 treatment) reduce binding specificity.

D) That the membrane actually plays an important role in helping to generate RNA selectivity (this is shown indirectly by loss of specificity in the Gag assembly mutants, but it is not clear that the membrane is required for Gag assembly).

E) That Gag proteins actually bind along the length of the viral RNA when the RNA-Gag-membrane complex forms.

This is not just a list of "extra" experiments to do – we consider them essential for validating the in vitro packaging system.

2) The authors should show tests of statistical significance throughout. Some of the changes shown appear to be rather modest, and it is important to know that they are statistically significant (or discover that they are not).

3) Molecular interpretations of the fluorescence data are very difficult. Of what the "Gag clusters" really consist remains unknown, and the fluorescence visualization in some cases is marginal. The lipid dye sometimes shows what GUV experts would interpret to be "junk" and does not track uniformly around the periphery of the GUV (e.g. Figure 1). While the bar graph showing that Gag mutants (esp. the two-fold WM mutant) are defective in clustering seems quite convincing (Figure 2), the actual GUV pictures used as examples (Figure 2 for wt, 2F for mutant) are not convincing at all. It is really hard to see much in these pictures and their quality should be improved.

Other points for the authors' consideration:

1) In Figure 1, why does addition of 10-fold excess of competitor RNA378 reduce Gag colocalization to 0% (I might have thought it would reduce it to 10%)?

2) In Figure 1, it would be helpful to highlight the splice donor site, since the lack of SHAPE reactivity at that site is used as evidence that the RNA is in the packageable conformation. The authors might even consider showing the alternative conformation with a SD hairpin to emphasize that their SHAPE data fit with one conformation and not the other.

3) In the subsection “SHAPE-based mapping of interactions between the HIV-1 5’UTR and Gag in solution and on membranes”: It was not clear why different protein:RNA ratios were used in the solution vs. GUV reactions.

4) A related question is how the authors know that most of the RNA is bound by Gag proteins under these conditions.

5) In the subsection “RNA selectivity depends upon intact CA domain lattice contacts”, last paragraph: It appears that the authors meant to refer to Figure 2 (not 2F).

6) Figure 6: One of the two RNAs seems to be missing its 3' tail (and the figure itself isn't labeled with a figure number).

7) The authors should probably discuss the point that the RNA structure from the Summers' laboratory that they use to interpret their SHAPE data is actually of the monomeric core packaging element. They argue, based upon the SHAPE data, that their RNA is dimeric even before it interacts with Gag, and this may well be true (though it should be shown more directly), but even if they're right it seems possible that the structures of the monomeric and dimeric core packaging RNA elements may differ in significant ways (beyond simply forming or not forming a DIS kissing loop).

---

## [Author Response]

*Substantive concerns to be addressed:*

*1) The authors argue that their work "will stand as the definitive in vitro study of HIV genome packaging".*

We admit that the word “definitive” might have sounded too strong. This word was only used in the cover letter and is absent from the manuscript.

*This is certainly a worthy goal, but to achieve it the authors need to do more to demonstrate that their system really does recapitulate all of the key aspects of specificity seen in the cellular context.*

*Specifically, the authors need to show:*

*A) Selective packaging of dimeric vs. mutant HIV RNAs (e.g., using DIS mutants that are shown experimentally not to dimerize and using other mutant(s) that have been shown to reduce packaging).*

*B) Selective packaging of unspliced RNAs (e.g., using mimics of spliced viral RNAs).*

These are excellent suggestions and very important points that concerned us as well. We address points A and B together in the new Figure 2. We generated the following mutant 5'UTR RNAs: a DIS-mutant monomeric construct, a mutation of packaging-critical guanines around three-way-1 junction, the combination of the monomerization and 3way1 mutations, and the first 345 nt of the spliced RNA env-1. We let these RNAs compete with fluorescent wild type 5'UTR for recruitment to Gag clusters on GUVs, and could measure that all of the mutant RNAs are weaker competitors than the wild type 5'UTR. We believe these data forcefully make the case that the in vitro system recapitulates the main features of biological HIV-1 genome packaging.

*C) That NC mutants (or AT-2 treatment) reduce binding specificity.*

The new data in Figure 1—figure supplement 1 show that either deletion of NC, or AT-2 treatment sharply reduce RNA recruitment, whereas a DMSO control still permits RNA recruitment to Gag clusters.

*D) That the membrane actually plays an important role in helping to generate RNA selectivity (this is shown indirectly by loss of specificity in the Gag assembly mutants, but it is not clear that the membrane is required for Gag assembly).*

Our data, and specifically the findings that specificity is lost in 2-fold and 6-fold axis mutants (Figure 4), strongly support our conclusion that Gag assembly is critical for specificity. The membrane is important in our model system as the platform on which all of the Gag assembly occurs, consistent with the accepted notion in the HIV field that the immature HIV-1 Gag lattice only assembles when Gag interacts with a membrane. The focus of HIV-1 budding research by our team is directed at reconstituting the most realistic possible in vitro model system for HIV-1 particle biogenesis and release, and this clearly requires the inclusion of membrane. It is possible to construct in vitro systems that assemble virus-like particles in the absence of membranes, notably the p6-deleted version of Gag studied by Rein and co-workers. It seems plausible to us that the system characterized by Rein might be able to selectively package RNA even in the absence of membranes, because that system is optimized for membrane-free assembly. Experimental tests of this seem to us to be far beyond the scope of this study and of our research program.

E) That Gag proteins actually bind along the length of the viral RNA when the RNA-Gag-membrane complex forms.

The best evidence we have to support binding along the length of the viral RNA is the gain in specificity relative to tRNA competition for the HIV1947 construct as compared to the 5’UTR construct (Figure 6). It would be difficult to explain this observation with other models, given the absence of additional known specific binding sites. We are not sure how this conclusion could be made stronger than it already is without bringing in some completely different technology (perhaps crosslinking coupled to deep sequencing, as already carried out by the Bieniasz group for Gag/RNA complexes isolated from cells) and making a major expansion of the scope of the study.

2) The authors should show tests of statistical significance throughout. Some of the changes shown appear to be rather modest, and it is important to know that they are statistically significant (or discover that they are not).

These have now been added throughout and support the significance of all the conclusions drawn previously.

*3) Molecular interpretations of the fluorescence data are very difficult. Of what the "Gag clusters" really consist remains unknown, and the fluorescence visualization in some cases is marginal. The lipid dye sometimes shows what GUV experts would interpret to be "junk" and does not track uniformly around the periphery of the GUV (e.g. Figure 1).*

The components the in vitro system are fully defined: PI(4,5)P_2_-containing GUVs, myristylated Gag, and in vitro transcribed RNA, so we do not feel it would be correct to state that the contents of the Gag clusters are “unknown”. With respect to the unevenly distributed membrane dye referred to as "junk": Mg^2+^ was added to the buffers in this study in order to ensure proper folding of the RNAs. We noticed that the presence of Mg^2+^ made the GUV morphology more variable, which is consistent with literature on the effect of Mg^2+^ on membranes containing anionic lipids. We now point this out in the text. The more variable GUV morphology does not influence the assignments of Gag and RNA clusters and causes no difficulty apart from a cosmetic one.

While the bar graph showing that Gag mutants (esp. the two-fold WM mutant) are defective in clustering seems quite convincing (Figure 2), the actual GUV pictures used as examples (Figure 2 for wt, 2F for mutant) are not convincing at all. It is really hard to see much in these pictures and their quality should be improved.

We have now increased their brightness in all figures by a uniform and equal amount. In addition, we marked the GUVs having diffuse Gag binding in the former Figure 2 (now Figure 3) with asterisks.

Other points for the authors' consideration:

1) In Figure 1, why does addition of 10-fold excess of competitor RNA378 reduce Gag colocalization to 0% (I might have thought it would reduce it to 10%)?

It is likely that the amount of fluorescent RNA at Gag clusters is reduced to 10%, not 0%. But our image analysis script makes a binary decision if a Gag cluster is positive for RNA based on an intensity threshold, and the reduction of the amount of fluorescent RNA by 90% apparently lowers the RNA fluorescence to below this threshold for all Gag clusters.

*2) In Figure 1, it would be helpful to highlight the splice donor site, since the lack of SHAPE reactivity at that site is used as evidence that the RNA is in the packageable conformation. The authors might even consider showing the alternative conformation with a SD hairpin to emphasize that their SHAPE data fit with one conformation and not the other.*

We assume that the reviewers referred to Figure 4, which is 5A in the revised manuscript. The region is now marked with "SD" and highlighted with a gray-shaded box, as mentioned in the revised text.

*3) In the subsection “SHAPE-based mapping of interactions between the HIV-1 5’UTR and Gag in solution and on membranes”: It was not clear why different protein:RNA ratios were used in the solution vs. GUV reactions.*

The SHAPE experiments on GUVs were conducted at lower RNA and protein concentrations. To ensure that little unbound RNA was present during those experiments (see below), we chose to use a higher protein:RNA ratio.

*4) A related question is how the authors know that most of the RNA is bound by Gag proteins under these conditions.*

In solution, the protein concentration of 7.2 µM is far in excess of the reported affinities between NC and 5'UTR (~17 nM according to Heng et al., 2012). For the studies on GUVs, we measured the RNA concentration in the GUV supernatant after a 5 min spin to pellet GUVs, and found that it was ~10-20% of added RNA, dependent on the presence of both Gag and GUVs. This is now stated in the Results section.

5) In the subsection “RNA selectivity depends upon intact CA domain lattice contacts”, last paragraph: It appears that the authors meant to refer to Figure 2 (not 2F).

Corrected.

*6) Figure 6: One of the two RNAs seems to be missing its 3' tail (and the figure itself isn't labeled with a figure number).*

We have corrected these omissions.

*7) The authors should probably discuss the point that the RNA structure from the Summers' laboratory that they use to interpret their SHAPE data is actually of the monomeric core packaging element. They argue, based upon the SHAPE data, that their RNA is dimeric even before it interacts with Gag, and this may well be true (though it should be shown more directly), but even if they're right it seems possible that the structures of the monomeric and dimeric core packaging RNA elements may differ in significant ways (beyond simply forming or not forming a DIS kissing loop).*

Heng et al. & Summers (2012) showed that the construct used was dimeric in the absence of Gag, and this is now referenced. The point that Figure 2 and Figure 5 show the monomeric structure is well-taken and that is now noted in the figure legend.